# Prediction and discrimination of skeletal muscle function by bioelectrical impedance vector analysis using a standing impedance analyzer in healthy Taiwanese adults

**Li-Wen Lee**[1], **Hsueh-Kuan Lu**[2], **Yu-Yawn Chen**[3,4], **Chung-Liang Lai**[5,6], **Lee-Ping Chu**[7], **Meng-Che Hsieh**[8], **Kuen-Chang Hsieh**[8,9]*

1 Department of Diagnostic Radiology, Chang Gung Memorial Hospital, Chiayi, Taiwan, 2 General Education Center, National Taiwan University of Sport, Taichung, Taiwan, 3 Department of Physical Education, National Taiwan University of Sport, Taichung, Taiwan, 4 Department of Food Science and Technology, National Taitung Junior College, Taitung, Taiwan, 5 Department of Physical Medicine and Rehabilitation, Puzi Hospital, Ministry of Health and Welfare, Chiayi, Taiwan, 6 Department of Occupational Therapy, Asia University, Taichung, Taiwan, 7 Department of Orthopedics, China Medical University Hospital, Taichung, Taiwan, 8 Research Center, Charder Electronic Co., Ltd, Taichung, Taiwan, 9 Fundamental Education Center, National Chin-Yi University of Technology, Taichung, Taiwan

* abaqus0927@yahoo.com.tw

## Abstract

### Background

Bioelectrical impedance vector analysis (BIVA) has been used for prediction of muscle performance. However, little is known about BIVA in Asian adults, and even less is known about using standing BIVA devices. Standing impedance analyzer allows quicker and more convenient way to gather data than conventional supine analyzer and is more suitable for clinical practice. This study aimed to investigate the relations between muscle function and BIVA parameters measured with a standing impedance analyzer in healthy Taiwanese adults.

### Methods

A total of 406 healthy subjects (age 34.5 ± 17.3 years, body mass index 24.1 ± 4.1 kg/m$^2$) were recruited for BIVA and handgrip strength (HGS) measurements. Impedance parameters, including resistance (R) and reactance (Xc), were measured and normalized to body size by dividing by height (H). The resulting phase angle (PhA) was calculated. HGS in the dominant, left, and right hands were referred to as HGS$_{DH}$, HGS$_{LH}$, and HGS$_{RH}$, respectively. All subjects were divided into 5 grades according to HGS.

### Results

Muscle strength in the dominant, right, and left arms was correlated with variables in the order of sex, weight, age, height, Xc/H, and R/H (all, p < 0.001). Using all 6 variables, the determination coefficients were 0.792, 0.782, and 0.745, respectively, whereas the standard errors of estimates were 56.89, 58.01, and 56.67 N for HGS$_{DH}$, HGS$_{LH}$, and HGS$_{RH}$,

**Data Availability Statement:** All relevant data are within the paper.

**Funding:** This work was supported by grant NSC100-2410-H-H-028-MY3 from the National Science Council of Taiwan and grant PG10601-0241 from Ministry of Health and Welfare of Taiwan.

**Competing interests:** The author's have read the journal's policy and have the following competing interests: Dr. Kuen-Chang Hsieh and a co-author Meng-Che Hsieh were employed by a commercial company, Charder Electronic Co., Ltd, during this study. This does not alter our adherence to PLOS ONE policies on sharing data and materials. There are no patents, products in development or marketed products associated with this research to declare.

respectively. HGS was positively correlated with PhA, and negatively correlated with Xc/H and R/H.

## Conclusions

BIVA parameters measured with a standing impedance analyzer and anthropometric variables can predict and discriminate muscle function with good performance in healthy Asian adults.

## Introduction

Skeletal muscle is the largest tissue in human body which accounts for 30% of body weight in women and greater than 40% of body weight in men [1, 2]. It controls physical activity through the generation of force and plays a major role in human health. The health of skeletal muscle is determined by its mass and function, and is regulated by skeletal muscle protein synthesis and breakdown [3]. Imbalance of the dynamic process of skeletal muscle protein metabolism in response to pathologic conditions and chronic disease may affect muscle mass and function [4]. Skeletal muscle function can be expressed in terms of muscle power, muscle strength, and local muscle endurance [5, 6]. Methods for muscle function evaluation include manual muscle testing, electrophysiological studies, and a handheld dynamometer. Handgrip strength (HGS) measurement using a dynamometer is a relatively inexpensive, portable, and simple method which provides information about overall muscle function [7]. The sex- and age-specific reference curves for HGS are well-established for healthy children and adults [8–10]. These reference curves provide normative values for physical fitness in general populations. Deviation of HGS values from the reference values may indicate disease [11] or the aging process [12, 13]. In the general population, a lower HGS is associated with higher risk of mortality and morbidity [11, 12].

Malnutrition is defined as a state resulting from lack of uptake or intake of nutrients. According to global consensus statements [14, 15], HGS is one of the four recommended types of measurements to assess nutritional state [14]. Studies have shown that lower HGS is associated with a longer length of hospital stay [16, 17] and higher mortality rate in critically ill patients [18]. HGS has also been used to monitor the outcome of nutritional intervention [19, 20]. Since muscle function may be altered prior to a change in muscular volume during disease progress or intervention, HGS is a more sensitive indicator to changes in nutritional status compared to body composition analysis [21, 22].

Bioelectrical impedance analysis (BIA) is a commonly used method for body composition analysis [23] and has been accepted as an option for estimating skeletal muscle mass [24, 25]. Another approach of BIA is bioelectrical impedance vector analysis (BIVA) introduced by Piccoli et al. [26]. Impedance analyzer measures tissue electrical properties such as resistance (R) and reactance (Xc) using an alternating current. Body fluids are highly conductive, and the resistance of the conductive fluids is defined as R [27]. Xc is the opposition to current flow due to the capacitive nature of cell membranes. Human cells are surrounded by phospholipid bilayers and act as an electrical insulator and capacitor. Therefore, Xc reflects the integrity of cell membranes, which is correlated with body cell mass [26, 28]. Phase angle (PhA) reflects the relationship of R and Xc, and is calculated as $PhA = \arctan(Xc/R)$ [29]. Impedance (Z) is a function of two impedance components, R and Xc, and is calculated as $Z^2 = R^2 + Xc^2$ [28]. If the two height-normalized impedance components (R/H and Xc/H) are plotted as a bivariate

vector in the RXc graph, the length of the vector (Mahalanobis distance) is related to hydration status [26]. Therefore, BIVA may be a one-stop shop solution for skeletal muscle health, providing information on both skeletal muscle mass and function. Currently, BIVA is a validated tool for assessing hydration and nutritional status [30] and has also been validated as a predictor for sports performance [31, 32] and muscular fitness [33].

Currently, BIVA measurements are typically made with subjects in a supine position, and data are acquired at the whole-body level. Moreover, there is very little published BIVA data on muscle function in healthy Asian adults. Indeed, standing impedance analyzer is a more attractive model in research and clinical settings owing to its simple and convenient measure. However, few studies have investigated BIVA data with subjects in standing position. Furthermore, there is a need to validate segmental BIVA for exploring muscle health in individual extremity. Consequently, the objective of this study was to investigate a modified standing impedance analyzer as a tool for skeletal muscle function at whole-body and segmental levels using HGS as a reference method in healthy Taiwanese adults.

## Materials and methods

This cross-sectional study was approved by the Institutional Review Board of the Jen-Ai Hospital (No. IRB-97-01). Written informed consent was obtained from all subjects. All experiments were conducted at the Jen-Ai Hospital in the Taichung, Taiwan between January 2016 and May 2017.

### Subjects

Subjects were recruited by community advertisements. Inclusion criteria were healthy Taiwanese adults 18 to 80 years of age. Exclusion criteria were individuals with a pacemaker, metal implants, limb deformities, upper limb neuropathies or arthropathies, those taking medications for chronic conditions and taking vitamin supplements long-term, and those with a history of alcohol abuse and systemic diseases, e.g., malignancy, diabetes, hypertension, hypo- or hyperthyroidism, cardiovascular disease. A total of 406 subjects who met the inclusion criteria and completed each measure were included in the final analysis.

### Study design

All subjects were asked to refrain from alcoholic beverages for at least 48 hours and avoid diuretics for 7 days prior to study. Female subjects were not scheduled during menstruation. On the test day, subjects were registered between 1 pm and 5 pm after fasting for 4 hours and were instructed to void, remove all objects which may affect the exam, and change into a light cotton gown prior to measurements.

### Anthropometric measurements

Height was measured to the nearest 0.1 cm using a mechanical device (Stadiometer, Holtain, Crosswell, Wales, UK), and weight was measured to the nearest 0.1 kg using an electronic scale (BC-418MA, Tanita Corporation, Tokyo, Japan) by skilled operators with subjects not wearing shoes. Technical errors for height and weight measurements were 0.021% and 0.520%, respectively. Body mass index (BMI) was calculated as weight (kg) divided by height squared (kg/m$^2$).

## Body composition measurements

Body composition measurements were acquired using a DXA scanner (GE, Lunar Prodigy, USA) by experienced radiology technicians. For the examination, subjects were placed supine on the scanning table with the upper limbs stretched and placed flatly on the side of the body, with the feet slightly parallel and the toes facing upwards. The total scan time was approximately 20 min. Regional cut lines were placed using enCore Version 7.0 software according to the manufacturer's protocol. The lean body mass and body fat percentage of the whole body, right arm, and left arm were obtained.

## Handgrip strength

A digital handgrip dynamometer (MG4800, Charder Electronical Co., Ltd., Taichung, Taiwan) was used measure HGS, after subjects were given verbal instructions and a brief demonstration. Subjects were instructed to stand upright with their shoulder adducted and neutrally rotated, elbow fully extended, and forearm and wrist neutrally positioned during the study. When correctly positioned, 3–5 second maximum grip strengths were obtained twice for each hand. No verbal encouragement was given during the test. The average values of the 2 trials in the dominant hand, right hand, and left hand were calculated, and represented as $HGS_{DH}$, $HGS_{RA}$, and $HGS_{LA}$, respectively.

To ensure the accuracy of the test, all testers were trained in the test procedures and calibration procedures, and instrument calibration data were recorded to ensure reproducibility of the test. All testers practiced the testing procedure in a subgroup of 35 subjects (age $35.2 \pm 12.3$ years, body weight $68.3 \pm 10.2$ kg, height $1.65 \pm 0.1$ m, BMI $23.9 \pm 3.3$ kg/m$^2$) prior to the study assessments. The test-retest reliability intra-class correlation coefficient (ICC) for HGS was r = 0.98 (95% CI: 0.93, 0.99). For criterion-related validity, the MG4800 dynamometer was validated against the standard Jamar dynamometer (J. A. Preston Corporation, Clifton, NJ). The results produced by the 2 devices were highly correlated (r = 0.954 by ICC), and strongly in agreement (bias = 12.0 N, limit of agreement = -58.5 to 85.5 N by Bland-Altman Plot).

## Impedance measurements

BIA measurements were carried out with the subject standing on a modified Quadscan 4000 (Bodystat Ltd, Doubles, Isle of Man, UK) with circuit switching switches and measuring lines [34]. The device was calibrated at the beginning of each day using a 500 ohm test resistor provided by the manufacturer, with R and Xc variations within 1% (R = 500 ± 5 ohm, Xc = 0 ± 5 ohm).The reliability and validity of the measuring device have been previously verified [34, 35].

The R, Xc, Z, and PhA for each subject were measured at a single frequency (50 kHz) with 3 modes: whole body (WB), right arm (RA), and the left arm (LA) modes (**Fig 1**). The method of BIVA was developed by Professor Antonio Piccoli in 1994 [26]. R and Xc were normalized to height (H), and expressed as R/H and Xc/H, respectively [26]. Then, R/H and Xc/H were used to plot a bivariate RXc graph. The 95% CIs, which represent the mean vector distribution, were calculated for the HGS measurements in the different groups. PhA was defined as arctan (Xc/R).

## Statistical analysis

We determined that this study needed a minimum sample size of 374 subjects for six estimate variables, using an effect size of 0.035 (f$^2$, medium), a 0.05 probability of error, and a power of 0.95 (1 − β error probability). All statistical analyses were performed using SPSS version 19.0

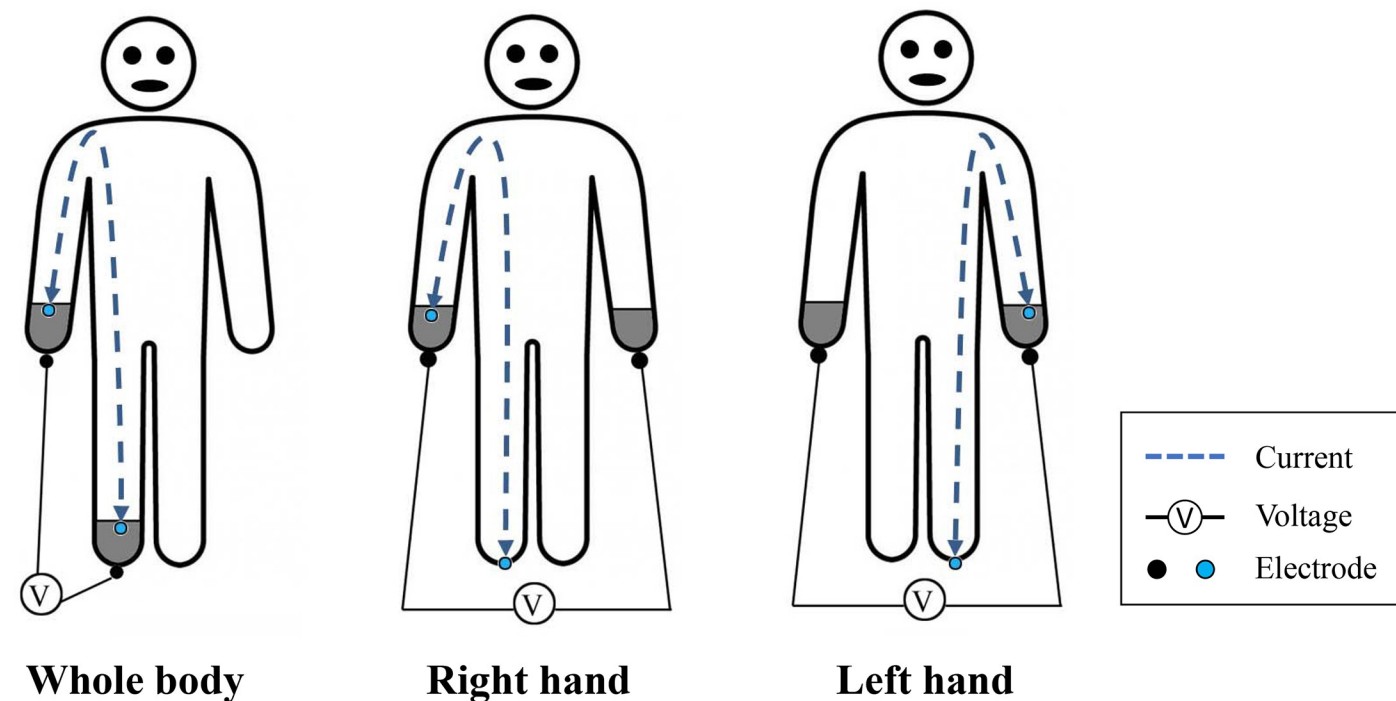

**Fig 1. Illustration of impedance measurement methods for the respective body parts.**

software (SPSS Inc., Chicago, IL, USA). Data were expressed as mean ± standard deviation (SD). The ICC was used to evaluate test-retest reliability. Repeated-measures ANOVA was used to test differences among group means between men and women. Pearson correlation (r) was used to assess the correlation between 2 variables. Stepwise regression analysis was used to fit possible regression models of muscle strength using sex, body weight, age, height, Xc/H, and R/H as independent variables ($F_{in} = 4.00$, $F_{out} = 3.99$). Vector plots and analyses were performed using BIVA Software 2002 [36]. Mean vector lengths of groups were tested using the Hotelling's T-squared test in the BIVA Software 2002. A value of $p < 0.05$ was considered to indicate statistical significance.

## Results

The demographic characteristics of the 406 subjects included in the study are presented in Table 1. Impedance parameters, with and without adjustment for height, are shown for the whole body and segmental levels. Mean age of the subjects was 34.5±17.3 years, and mean BMI was 24.1±4.1 kg/m$^2$. Of the 406 subjects, 235 (58%) were men. Anthropometric indices, HSG, and impedance parameters were significantly greater for men compared to women, except for age and BMI. The majority of subjects were right-handed (94.3%). The mean HGS of the dominant hand was 366.8 ± 108.0 N for men and 251.0 ± 58.5 N for women. The correlation coefficients between muscle mass and function in the dominant hand, right arm, and left arm were 0.866, 0.810, and 0.823, respectively (all, $p < 0.001$).

Possible associations between BIVA variables PhA, R/H, and Xc/H obtained using whole body and segmental modes were examined (Table 2). All BIVA variables obtained from whole body and segmental modes were very strongly correlated in both sexes (r = 0.910–0.985), except for PhA in males (r = 0.852–0.897). The best association was found with R/H

**Table 1. Subject demographics.**

| | Total (n = 406) | Female (n = 171) | Male (n = 235) |
|---|---|---|---|
| Age (year) | 34.5±17.3 (18.7, 79.0) | 36.9±18.7 (18.8, 79.8) | 32.8±15.4 (18.7, 79.8) |
| Height (m) | 1.68±0.10 (1.45, 1.97) | 1.61±0.06 (1.46, 1.74) | 1.74±0.08 (152, 197)** |
| Weight (kg) | 67.9±14.1 (42.0, 120.0) | 60.8±11.9 (42.0, 106.0) | 75.3±12.2 (45.0, 120.0) ** |
| BMI (kg/m$^2$) | 24.1±4.1 (16.2, 39.9) | 23.5±4.6 (16.2, 38.0) | 24.6±3.6 (16.3, 39.9) |
| HGS$_{DH}$ (N) | 366.8±108.0 (86.3, 601.7) | 251.0±58.5 (127.5, 402.5) | 482.7±82.8 (188.7, 721.0) ** |
| HGS$_{RH}$ (N) | 363.5±108.2 (86.3, 601.7) | 249.5±58.5 (127.5, 388.2) | 482.7±82.8 (188.7, 721.0) ** |
| HGS$_{LH}$ (N) | 337.8±99.1 (78.2, 552.3) | 231.4±59.3 (112.2, 402.5) | 437.7±77.7 (178.9, 638.6) ** |
| Z$_{WB}$ (ohm) | 559.6±99.4 (376.7, 908.6) | 650.0±96.0 (476.7, 908.6) | 514.0±63.7 (376.7, 671.7) ** |
| Z$_{RA}$ (ohm) | 312.5±68.0 (203.5, 509.4) | 379.4±61.8 (263.6, 509.4) | 278.8±40.9 (203.5, 389.9) ** |
| Z$_{LA}$ (ohm) | 319.4±70.7 (194.9, 522.3) | 386.8±64.7 (268.2, 522.2) | 285.5±44.6 (194.9, 423.7) ** |
| R$_{WB}$ | 556.5±99.4 (374.6, 905.8) | 647.8±96.0 (474.2, 905.8) | 510.7±63.4 (374.6, 668.1) ** |
| R$_{RA}$ | 311.0±68.1 (202.5, 508.0) | 378.1±61.8 (262.5, 508.0) | 277.3±40.7 (202.5, 388.4) ** |
| R$_{LA}$ | 318.0±70.7 (193.9, 520.9) | 385.5±64.8 (267.1, 520.9) | 284.0±44.5 (193.9, 422.2) ** |
| Xc$_{WB}$ | 58.1±7.7 (40.1, 79.6) | 58.4±7.2 (44.9, 79.6) | 57.9±8.0 (40.1, 78.9) ** |
| Xc$_{RA}$ | 30.1±4.3 (19.9, 40.2) | 31.7±3.9 (23.4, 40.0) | 29.3±4.3 (19.9, 40.2) ** |
| Xc$_{LA}$ | 29.9±4.2 (19.4, 41.2) | 31.0±3.9 (23.0, 40.3) | 29.3±4.2 (19.4, 41.2) ** |
| R$_{WB}$/H (ohm/m) | 332.1±68.2 (214.6, 595.9) | 401.9±62.9 (283.3, 595.9) | 297.1±36.7 (214.6, 370.7) ** |
| R$_{RA}$/H (ohm/m) | 185.8±45.8 (116.1, 334.2) | 236.4±39.9 (158.1, 334.2) | 161.3±23.6 (116.1, 218.2) ** |
| R$_{LA}$/H (ohm/m) | 189.9±47.2 (111.1, 342.7) | 239.1±41.6 (160.9, 342.7) | 165.2±25.6 (111.1, 237.2) ** |
| Xc$_{WB}$/H (ohm/m) | 34.6±5.0 (22.5, 48.2) | 36.3±4.9 (26.4, 48.2) | 33.7±4.8 (22.5, 46.5) ** |
| Xc$_{RA}$ /H (ohm/m) | 18.0±2.9 (11.1, 25.1) | 19.7±2.7 (13.8, 25.1) | 17.1±2.6 (11.1, 24.5) ** |
| Xc$_{LA}$ /H(ohm/m) | 17.8±2.8 (11.1, 25.1) | 19.2±2.7 (14.1, 24.4) | 17.0±2.5 (11.1, 25.1) ** |
| PhA$_{WB}$ (˚) | 6.0±0.8 (3.5, 7.2) | 5.2±0.6 (3.9, 6.4) | 6.2±0.5 (4.9, 7.2) ** |
| PhA$_{RA}$ (˚) | 5.5±0.9 (4.1, 8.6) | 4.8±0.6 (3.5, 6.1) | 6.0±0.7 (4.5, 8.2) ** |
| PhA$_{LA}$ (˚) | 5.4±0.9 (3.8, 8.2) | 4.6±0.7 (3.5, 6.1) | 5.9±0.7 (4.9, 8.6) ** |
| Lean$_{WB}$ (kg) | 48.0±11.6 (24.7, 82.3) | 37.2±4.7 (24.7, 53.4) | 55.8±8.0 (33.4, 82.2) ** |
| Lean$_{RA}$ (kg) | 2.7±0.9 (1.1, 5.2) | 1.8±0.4 (1.1, 3.2) | 3.3±0.6 (1.6, 5.1) ** |
| Lean$_{LA}$ (kg) | 2.6±0.9 (1.1, 4.9) | 1.8±0.5 (1.1, 3.4) | 3.2±0.6 (1.7, 4.9) ** |
| BF% (%) | 25.3±11.4 (5.1, 54.3) | 33.1±9.4 (10.4, 54.3) | 19.6±9.2 (5.1, 40.4) ** |

[a] Data are expressed as mean ± standard deviation (min, max).

[b] HGS, hand grip strength; Z, impedance; R, resistance; Xc, reactance; R/H, resistance standardized for height; Xc/H, reactance standardized for height; PhA, phase angle; Lean, lean body mass; BF%, percentage body fat

*, $p < 0.05$

**, $p < 0.001$.

[c] Subscript DH, RH, LH, WB, RA, and LA denote dominant head, right hand, left hand, whole body, right arm and left arm, respectively.

(r = 0.943–0.985). In general, the correlations between whole body and right arm modes were better than the correlations between whole body and left arm modes for all the BIVA variables.

The results of the multiple regression analyses for HGS using basic indices and height-adjusted BIVA variables (R/H and Xc/H) as predictors are shown in Table 3. The variables were included in the stepwise regression analysis in the following order: sex, weight, age, height, Xc/H, and R/H. Model 1 was the regression model to predict HGS$_{DH}$ from basic indices and whole-body mode BIVA parameters. Model 2 was the regression model to predict HGS$_{RH}$ based on basic indices and right hand mode BIVA parameters. Model 3 was the regression model to predict HGS$_{LH}$ from basic indices and left hand mode BIVA parameters. The

**Table 2. Regression analysis of impedance parameters using whole body and segmental modes.**

| | Dependent Variable | Independent Variable | Intercept | Coefficient | $r^2$ | SEE |
|---|---|---|---|---|---|---|
| Total (n = 406) | $Ph_{WB}$ | $PhA_{RA}$ | 0.686±0.127** | 0.939±0.022** | 0.960 | 0.240 |
| | | $PhA_{LA}$ | 1.217±0.142** | 0.871±0.025** | 0.941 | 0.290 |
| | $R_{WB}/H$ | $R_{RA}/H$ | 64.055±3.784** | 1.447±0.020** | 0.985 | 11.885 |
| | | $R_{LA}/H$ | 66.924±4.291** | 1.400±0.022** | 0.981 | 13.560 |
| | $Xc_{WB}/H$ | $Xc_{RA}/H$ | 5.434±0.923** | 1.623±0.051** | 0.962 | 0.755 |
| | | $Xc_{LA}/H$ | 1.098±1.001** | 0.929±0.056** | 0.931 | 1.829 |
| Female (n = 171) | $Ph_{WB}$ | $PhA_{RA}$ | 0.744±0.228** | 0.915±0.047** | 0.939 | 0.210 |
| | | $PhA_{LA}$ | 1.557±0.219** | 0.777±0.046** | 0.920 | 0.241 |
| | $R_{WB}/H$ | $R_{RA}/H$ | 40.748±11.247** | 1.539±0.047** | 0.976 | 13.481 |
| | | $R_{LA}/H$ | 50.328±12.174** | 1.469±0.050** | 0.971 | 14.901 |
| | $Xc_{WB}/H$ | $Xc_{RA}/H$ | 2.154±1.859* | 1.734±0.094** | 0.933 | 1.779 |
| | | $Xc_{LA}/H$ | 3.714±2.099* | 1.693±0.108** | 0.910 | 2.051 |
| Male (n = 235) | $Ph_{WB}$ | $PhA_{RA}$ | 1.339±0.252** | 0.838±0.040** | 0.897 | 0.243 |
| | | $PhA_{LA}$ | 1.992±0.276** | 0.751±0.045** | 0.852 | 0.288 |
| | $R_{WB}/H$ | $R_{RA}/H$ | 65.642±6.662** | 1.443±0.042** | 0.959 | 10.731 |
| | | $R_{LA}/H$ | 79.702±7.506** | 1.322±0.046** | 0.943 | 12.650 |
| | $Xc_{WB}/H$ | $Xc_{RA}/H$ | 3.243±1.006** | 1.784±0.058** | 0.948 | 1.537 |
| | | $Xc_{LA}/H$ | 3.443±1.166** | 1.776±0.068** | 0.932 | 1.762 |

Data are presented as regression coefficient estimate ± standard error of estimate.

[b] $r^2$, coefficient of determination.; H, height; PhA, phase angle; R, resistance; Xc, reactance; R/H, resistance standardized for height; Xc/H, reactance standardized for height; Subscript RA, LA, WB = right arm, left arm, whole body, respectively.

*, $p < 0.05$

**, $p < 0.001$.

variance inflation factor (VIF) values were all < 10 (range 1.25–8.26), indicating no multicollinearity. The correlation coefficients between HGS and Xc/H in the whole body, right arm, and left arm modes were 0.663, 0.690, and 0.651, respectively. The correlation coefficients between muscle strength and R/H in whole body, right arm, and left arm modes were 0.773, 0.775 and 0.747, respectively. In general, Xc/H was a better predictor for muscle strength than R/H in all 3 modes.

The subjects were divided into 5 equal groups (group I to V) depending on their HGS level, with group V representing the lowest quintile. Graphical comparisons of impedance vectors and confidence ellipses are shown in Fig 2: $HGS_{DH}$ using whole body mode (Fig 2A); $HGS_{RH}$ using right arm impedance measuring mode (Fig 2B); $HGS_{LH}$ using left arm mode (Fig 2C). A significant displacement of the vector was observed between groups with increasing HGS in all 3 models (p = 0.0001–0.0112 in the whole body model, p = 0.0001–0.002 in the right arm model, p = 0.0001–0.0245 in the left arm model). With increasing level of HGS, a decreasing PhA was also noted in all 3 models (**Fig 2**).

## Discussion

The results of this study showed that HGS can be predicted by BIVA parameters of the same limb, and the whole body using a modified standing impedance analyzer. Standing BIA analyzers have attracted a growing interest due to their convenience; however, there are concerns about impedance variability due to fluid shift toward the leg during the day [37, 38]. Our study provides evidence that standing BIVA can be used to predict and discriminate muscle function

**Table 3. Multiple regression analyses for predicting handgrip strength in the dominant, right, and left hands.**

| Sex | Weight | Age | H | Xc/H | R/H | Intercept | SEE | r² | VIF | β |
|---|---|---|---|---|---|---|---|---|---|---|
| Cumulative dependent variables used in model | | | | | | | | | | |
| Whole body mode for predicting HGS$_{DH}$ | | | | | | | | | | |
| 192.16±10.66** | - | - | - | - | - | 294.83± 7.55** | 74.79 | 0.625 | 3.45 | 0.42 |
| 149.40±10.89** | 2.96±0.39** | - | - | | | 114.72± 24.39** | 65.67 | 0.712 | 2.82 | 0.23 |
| 141.01±10.04** | 3.08±0.35** | -1.91±0.30** | - | - | - | 166.37±23.75** | 60.00 | 0.761 | 1.48 | -0.11 |
| 118.38±11.97** | 2.60±0.37** | -1.55±0.31** | 2.18±0.66** | - | - | -165.73±103.70 | 58.53 | 0.774 | 2.66 | 0.22 |
| 126.38±13.79** | 2.82±0.42** | -1.50±0.32** | 2.23±0.66** | 0.78±0.67* | - | -232.56±118.42 | 57.47 | 0.784 | 6.69 | 0.29 |
| 102.90±15.06** | 1.95±0.48** | -.96±0.35** | 2.65±0.66** | 3.42±1.01* | -0.66±0.19** | -1 48.70±117.77 | 56.89 | 0.792 | 7.26 | -0.38 |
| Right arm mode for predicting HGS$_{RH}$ | | | | | | | | | | |
| 191.68±10.81** | - | - | - | - | - | 291.48±7.66** | 75.82 | 0.617 | 3.49 | 0.40 |
| 150.38±11.19** | 2.86±0.39** | - | - | - | - | 117.54±25.07** | 67.51 | 0.698 | 2.49 | 0.18 |
| 141.73±10.31** | 2.98±0.36** | -1.97±0.31** | - | - | - | 170.86±24.39** | 61.62 | 0.750 | 1.25 | -0.14 |
| 116.69±12.24** | 2.45±0.38** | -1.58±0.32** | 2.41±0.68** | - | - | -196.57±106.03 | 59.84 | 0.765 | 2.64 | 0.23 |
| 113.27±15.29** | 2.39±0.41** | -1.57±0.32** | 2.41±0.68** | 0.28±0.74* | - | -181.21±113.88 | 59.97 | 0.766 | 5.60 | 0.18 |
| 96.44±15.45** | 1.51±0.46** | -1.22±0.33** | 2.83±0.67** | 2.26±.98* | -0.94±0.25** | -101.23±112.17 | 58.01 | 0.782 | 7.21 | -0.34 |
| Left arm mode for predicting HGS$_{LH}$ | | | | | | | | | | |
| 170.26±12.42** | - | - | - | - | - | 267.28±7.10** | 70.29 | 0.773 | 3.55 | 0.42 |
| 134.51±10.56** | 2.47±0.37** | - | - | - | - | 116.71±23.64** | 63.65 | 0.671 | 2.58 | 0.18 |
| 126.66±9.79** | 2.59±0.34** | -1.78±0.29** | - | - | - | 165.05±23.18** | 58.55 | 0.719 | 1.25 | -0.16 |
| 111.33±.998** | 2.26±0.37** | -1.54±0.32** | 1.48±0.66* | - | - | -59.87±102.26 | 57.95 | 0.730 | 2.71 | 0.18 |
| 109.44±14.69** | 2.23±0.40** | -1.55±0.31** | 1.48±0.66** | -0.15±0.67* | - | -52.12±108.69 | 58.09 | 0.730 | 4.93 | 0.16 |
| 92.39±15.21** | 1.43±0.46* | -1.25±0.32** | 1.98±0.66* | 1.75±0.87* | -0.76±0.23** | -102.47±107.26 | 56.67 | 0.745 | 6.76 | -0.31 |

[a] Data are presented as regression coefficient estimate ± standard error of estimate.

[b] SEE, standard error of estimate; r², coefficient of determination; VIF, variance inflation factor; β, standardized coefficient.

[c] Using all 6 variables, the determination coefficients were 0.792, 0.782, and 0.745, respectively, whereas the SEE were 56.89, 58.01, and 56.67 N for the dominant, right, and left arms, respectively.

[d] *, $p < 0.05$

**, $p < 0.001$.

in healthy adults. BIVA references for the healthy Asian adult population are limited because most BIVA studies have been conducted with Caucasian subjects [26, 31, 39–43]. The current study fills this knowledge gap by providing references ranges for the Asian population.

Compared to body composition analysis, vector analysis uses height-adjusted raw impedance components (R/H and Xc/H), and involves fewer assumptions and is free of prediction equations for total body water or fat free mass [32]. Therefore, vector analysis should exhibit less error than body composition analysis using an impedance analyzer, making it a more valuable tool. However, BIVA approach involves the use of raw bioelectrical impedance parameters and thus it is still sensitive to variability in tissue electric properties, such as body position, hydration status, electrolyte concentration, exercise, skin temperature and phase of menstrual cycle [28, 44–46]. This study was performed to test the potential application of BIVA in assessing skeletal muscle function and had the advantage of testing on healthy subjects under a careful control of hydration status.

The most common application of bioelectric impedance measure is whole body mode, which measures the total body electrical parameters with electrodes placed on the ipsilateral arm and foot. Whole body BIVA result is well-known to correlate with many diseases, such as

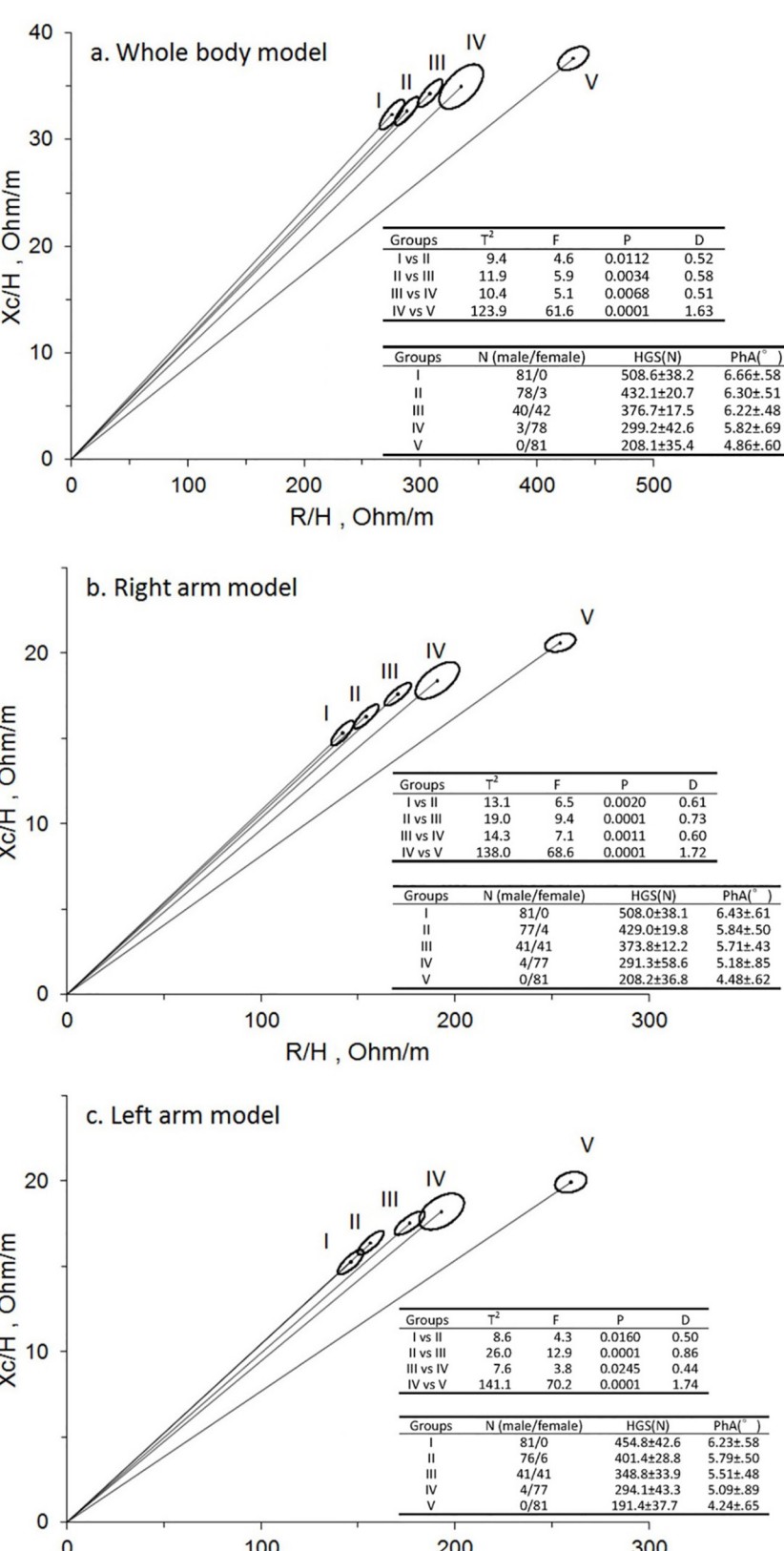

**Fig 2. The RXc graph with 95% confidence ellipses for the handgrip strength quintile groups.** (a) Whole body model (b) Right hand model (c) Left hand model. [a] Hand grip strength (HGS) is presented as mean ± standard deviation. [b] R, resistance; H, height; Xc, reactance; T[2], Hotelling's T-squared test; D, Mahalanobis distance; N, newton.

renal diseases, critically ill patients, obesity, sarcopenia, and cachexia [40, 41, 47–49]. However, whole body mode regards the body as a uniform cylinder and this assumption is not entirely correct as the human body has a complex shape [28]. Furthermore, this method cannot provide information about body segment individually, limiting its application.

Segmental body measure of bioelectric impedance is a less common application, which can be acquired by two methods. For the bioelectric impedance of the arm, segmental impedance components can be acquired using two pairs of electrodes attached to the ipsilateral hand and shoulder [49]. The distance between receiving electrodes is then used for adjusting the impedance parameters. This method has been used to assess the skeletal muscle of right arm in Alzheimer's disease and healthy control, showing a lower phase angles and longer specific vectors in patients with Alzheimer's disease [50]. However, this approach involves complicated process for electrode placement and is less practical in large scale study. Another segmental approach is proposed by Kushner R.F. [28] who made the assumptions that the human body is composed of five cylinders (two upper limb cylinders, one trunk cylinder and two lower lib cylinders) and tissue electric properties in each cylinder can be directly measured. For BIVA of the arm, impedance components are measured with electrodes placed on the ipsilateral arm and foot, and voltage electrodes on both hands [28, 51], as done in our study. With this method, the measured values of Xc and R are divided by standing height, which assumes a fixed proportion of limb length to body height. This approach is simple and convenient but less discussed.

Maximum handgrip force is mainly determined by the muscle function of the upper limb; therefore, HSG should be associated with BIVA of the ipsilateral limb rather than BIVA of the contralateral limb or the whole body BIVA. Interestingly, whole body Xc/H and R/H showed similar performance in predicting HGS compared to segmental Xc/H and R/H of the same limb. This may due to the very strong correlation between whole body and segmental Xc/H and R/H in our healthy subjects. However, our results may not be applicable to individuals with diseases, or the general, non-Asian population. Further research is required to explore the used direct measurement of segmental BIVA for the evaluation of regional muscle function.

In this study, a significant migration of the mean vectors with increasing HGS was observed due to decreased Xc/H and R/H in the healthy adults. A similar finding was reported in a study of healthy young adults by Rodriguez-Rodriguez et al [33]. A study of in inpatient subjects showed a different trend for the vector shift; the average vector displaced with increasing HSG due to decreased R/H, but increased Xc/H [29]. Interestingly, our study showed an increase in PhA with increasing HSG in healthy adults using whole body, right limb, and left limb modes, which is consistent with previous studies of healthy young adult [33] and inpatient subjects [29]. Moreover, PhA has been validated as a good predictor of nutritional and functional status [52–54].

In this study, we developed a regression model for HGS using basic indices and BIVA components (Xc/H and R/H). Similar study has been performed by Norman et al. in hospitalized European patients [29]. Their study included the same independent variables in the regression analysis for muscle function as ours, but our models performed better with higher R-squared values ($r^2 = 0.745$–792) than theirs ($r^2 = 0.708$). Additionally, the exact order the variables were entered into the equation were different in the 2 studies. In the Norman et al. study, the order was height, age, sex, weight, Xc/H, and R/H; whereas, in our study the order was sex, weight,

age, height, Xc/H, and R/H. Although adjusted BIVA components were entered into the models later than the anthropometric indices, correlations between HGS with Xc/H and with R/H were moderate to strong in both studies.

There are limitations of this study that should be considered. First, our study was conducted with healthy Asian adults, and thus the results may not be applicable to different populations or individuals with diseases. Second, migration of the tolerate ellipses in the RXc plots were correlated with group-level differences, and the BIVA method may not be sensitive enough to evaluate vector shifts at the individual level. Third, the segmental BIVA in this study was not measured with electrodes placed on the upper limbs. However, this modified method is more convenient, and yielded good results. Fourth, HGS measured by hand dynamometer may be affected by instrument-, operator- and subject-related errors.

## Conclusions

Our study showed that BIVA parameters measured by a standing impedance analyzer and anthropometric variables can predict muscle function as measured by HGS with good performance in healthy Asian adults. Our results may facilitate clinical applications of standing BIVA technology in assessing skeletal muscle function.

## Author Contributions

**Conceptualization:** Li-Wen Lee, Hsueh-Kuan Lu, Yu-Yawn Chen, Chung-Liang Lai, Kuen-Chang Hsieh.

**Formal analysis:** Li-Wen Lee, Kuen-Chang Hsieh.

**Funding acquisition:** Yu-Yawn Chen, Chung-Liang Lai.

**Investigation:** Li-Wen Lee, Lee-Ping Chu.

**Methodology:** Li-Wen Lee, Meng-Che Hsieh.

**Project administration:** Chung-Liang Lai.

**Resources:** Hsueh-Kuan Lu, Yu-Yawn Chen, Chung-Liang Lai, Kuen-Chang Hsieh.

**Supervision:** Kuen-Chang Hsieh.

**Validation:** Li-Wen Lee, Hsueh-Kuan Lu, Chung-Liang Lai, Meng-Che Hsieh, Kuen-Chang Hsieh.

**Visualization:** Li-Wen Lee, Yu-Yawn Chen, Lee-Ping Chu, Kuen-Chang Hsieh.

**Writing – original draft:** Li-Wen Lee, Kuen-Chang Hsieh.

**Writing – review & editing:** Li-Wen Lee, Hsueh-Kuan Lu, Yu-Yawn Chen, Chung-Liang Lai, Kuen-Chang Hsieh.

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
