## [Decision Letter · Decision Letter 0]

1 May 2020

PONE-D-20-07678

Prediction and discrimination of skeletal muscle function by bioelectrical impedance vector analysis using a standing impedance analyzer in healthy Taiwanese adults

PLOS ONE

Dear Dr. Hsieh,

Thank you for submitting your manuscript to PLOS ONE. After careful consideration, we feel that it has merit but does not fully meet PLOS ONE’s publication criteria as it currently stands. Therefore, we invite you to submit a revised version of the manuscript that addresses the points raised during the review process.

Minor and major weaknesses have been identified that need to be addressed. It would be particularly interesting for the authors to address those highlighted by the reviewer 2. The focus should be on improving the discussion of the study according to the requirements expressed.

We would appreciate receiving your revised manuscript by Jun 15 2020 11:59PM. To enhance the reproducibility of your results, we recommend that if applicable you deposit your laboratory protocols in protocols.io, where a protocol can be assigned its own identifier (DOI) such that it can be cited independently in the future. For instructions see: http://journals.plos.org/plosone/s/submission-guidelines#loc-laboratory-protocols

We look forward to receiving your revised manuscript.

Kind regards,

Jose M. Moran

Academic Editor

PLOS ONE

Additional Editor Comments:

Minor and major weaknesses have been identified that need to be addressed. It would be particularly interesting for the authors to address those highlighted by the reviewer 2. The focus should be on improving the discussion of the study according to the requirements expressed.

Journal Requirements:

'Competing Interests: Two of the authors (KCH) and (MCH) are employed by Charder Electronic Co, Ltd. This company did not provide KCH and MCH financial support in executing this study. Nor did the company have any additional role in the research funding, study design, data collection and analysis, decision to publish, or preparation of the manuscript. There are no patents, products in development nor marketed products to be declared. The other authors declare no conflict of interest.'

Reviewers' comments:

Reviewer's Responses to Questions

**Comments to the Author**

1. Is the manuscript technically sound, and do the data support the conclusions?

Reviewer #1: Yes

Reviewer #2: Yes

2. Has the statistical analysis been performed appropriately and rigorously? 

Reviewer #1: Yes

Reviewer #2: Yes

3. Have the authors made all data underlying the findings in their manuscript fully available?

Reviewer #1: Yes

Reviewer #2: Yes

4. Is the manuscript presented in an intelligible fashion and written in standard English?

Reviewer #1: No

Reviewer #2: Yes

5. Review Comments to the Author

Reviewer #1: 59 - Insert period.

69 - Insert 'the' before general population.

70-79 - Specific reported data from other studies is not necessary to include in the Introduction. This can be distracting to the reader. I suggest only including your study data reported in the Results section.

89 - Insert 'an' before alternating.

227-228 - Include parenthesis when describing 20% for group I and V, or not at all. Regardless, stay consistent.

283-284 - Which goodness of fit test was used.

Reviewer #2: With this study authors aimed to determine if bioelectrical impedance vector analysis assessment can predict muscle function in a specific healthy adult population. The methods used are appropriate but some details should be added to increase the completeness of the manuscript as well as additional discussion points that will aid in reader comprehension and interpretation.

Introduction should include a better justification of the study rational and clarify the actual knowledge about bioelectrical impedance vector analysis to discriminate muscle function as well as BIVA assessment positions.

Discussion should include appraisal with other methodologies and results used in previous studies. It also should be discussed how the circumstances BIVA was tested (refrain from alcoholic beverages for at least 48 hours, avoid diuretics for 7 days fast for 4 hours prior to study, and not asses females during menstruation) can interfere with the results.

Specific Comments:

Line

55 “Skeletal muscle is the largest organ in human body”: please review the statement and add the bibliographic reference

59 “and chronic disease may affect muscle mass and function [2] Muscle mass can be indirectly”: add punctuation mark period

62 “local muscle endurance [3].” Please verify the reference used

69 “In general population, a lower HSG is”: Correct HGS

81, 82 “HGS is one of the recommended tools for assessing nutritional state [13].”: Consider altering to: HGS is one of the four recommended types of measurements to assess nutritional state. [13].

226 “strsu3ength than R/H in all 3 modes.”: Change to: strength

258, 259 “Alternatively, segmental BIVA components may be measured with electrodes placed on the ipsilateral arm and foot, and voltage electrodes on both hands..”: Please add reference

Table 1: verify legend

Figure 1: add figure title and verify legend

6. PLOS authors have the option to publish the peer review history of their article (what does this mean?). If published, this will include your full peer review and any attached files.

Reviewer #1: No

Reviewer #2: No

---

## [Author Response · Author response to Decision Letter 0]

13 May 2020

Comments from reviewers:

Reviewer #1

59 - Insert period.

[Answer]: 

Thanks for pointing out the error. We have corrected the mistake as below:

(line 59) and chronic disease may affect muscle mass and function [2].

69 - Insert 'the' before general population.

[Answer]: 

Thanks for pointing out the error. We have added the article “the” before the noun as below:

(line 69) In “the” general population, a lower HSG is……..

70-79 - Specific reported data from other studies is not necessary to include in the Introduction. This can be distracting to the reader. I suggest only including your study data reported in the Results section.

[Answer]: 

Thanks for the suggestion. We have deleted line 70-79 in the revised manuscript. In addition, the rest of 2nd paragraph was merged with the 1st paragraph. 

89 - Insert 'an' before alternating.

[Answer]: 

Thanks for pointing out the error. We have corrected it in the revised manuscript as below:

(line 89) properties such as resistance (R) and reactance (Xc) using “an” alternating current.

227-228 - Include parenthesis when describing 20% for group I and V, or not at all. Regardless, stay consistent.

[Answer]: 

The sentence has been rephased as below:

(line 227-229) The subjects were divided into 5 equal groups (group I to V) depending on their HGS level, with group V representing the lowest quintile. 

283-284 - Which goodness of fit test was used.

[Answer]: 

To make the sentence clear, we have change “goodness of fit” to R-squared values in the revised manuscript as below:

Compared to the regression model for HGS developed using inpatient subjects by Norman et al. [25], the independent variables included in our regression analysis for muscle function were the same, but our models exhibited better R-squared values (r2 = 0.745-792) than theirs (r2 = 0.708).

 

Reviewer #2

With this study authors aimed to determine if bioelectrical impedance vector analysis assessment can predict muscle function in a specific healthy adult population. The methods used are appropriate but some details should be added to increase the completeness of the manuscript as well as additional discussion points that will aid in reader comprehension and interpretation.

Introduction should include a better justification of the study rationale and clarify the actual knowledge about bioelectrical impedance vector analysis to discriminate muscle function as well as BIVA assessment positions.

Discussion should include appraisal with other methodologies and results used in previous studies. It also should be discussed how the circumstances BIVA was tested (refrain from alcoholic beverages for at least 48 hours, avoid diuretics for 7 days fast for 4 hours prior to study, and not asses females during menstruation) can interfere with the results.

[Answer]:

We thank the reviewer for the suggestions. We have revised the introduction and discussion sections in accordance with the comments from the reviewer. The main changes in the introduction can be found in the last two paragraphs of then section. 

The main reason for the subject preparation before BIVA such as refrain from alcoholic beverages for at least 48 hours, avoid diuretics for 7 days fast for 4 hours prior to study, and not asses females during menstruation is to have a good control of hydration status of the subject as variation in hydration status may affect the accuracy and precision of electric properties of biological tissues. We have done a literature review and summarized the methodologies in the end of the 2nd paragraph in discussion as below:

However, BIVA approach still involves the use of raw bioelectrical impedance parameters and is sensitive to physiological changes associated with the tissue conductivity. Factors known to introduce variability in tissue electric properties include body position, hydration status, electrolyte concentration, exercise, skin temperature and phase of menstrual cycle [28, 44-46]. This study was performed to test the potential application of BIVA in assessing skeletal muscle function and had the advantage of testing on healthy subjects under a careful control of hydration status.

Also, we have included appraisal with the other BIVA methodologies and results in the 3rd and 4th paragraphs of discussion in the revised manuscript as below: 

The most common application of bioelectric impedance measure is whole body mode, which measures the total body electrical parameters with electrodes placed on the ipsilateral arm and foot. Whole body BIVA result is well-known to correlate with many diseases, such as renal diseases, critically ill patients, obesity, sarcopenia, and cachexia [40, 41, 48-50]. However, whole body mode regards the body as a uniform cylinder and this assumption is not entirely correct as the human body has a complex shape [28]. Furthermore, this method cannot provide information about body segment individually, limiting its application.

Segmental body measure of bioelectric impedance is a less common application, which can be acquired by two methods. For the bioelectric impedance of the arm, segmental impedance components can be acquired using two pairs of electrodes attached to the ipsilateral hand and shoulder [50]. The distance between receiving electrodes is then used for adjusting the impedance parameters. This method has been used to assess the skeletal muscle of right arm in Alzheimer’s disease and healthy control, showing a lower phase angles and longer specific vectors in patients with Alzheimer’s disease [51]. However, this approach involves complicated process for electrode placement and is less practical in large scale study. Another segmental approach is proposed by Kushner R.F. [28] who made the assumptions that the human body is composed of five cylinders (two upper limb cylinders, one trunk cylinder and two lower lib cylinders) and tissue electric properties in each cylinder can be directly measured. For BIVA of the arm, impedance components are measured with electrodes placed on the ipsilateral arm and foot, and voltage electrodes on both hands [28, 52], as done in our study. With this method, the measured values of Xc and R are divided by standing height, which assumes a fixed proportion of limb length to body height. This approach is simple and convenient but less discussed.

Specific Comments:

Line

55 “Skeletal muscle is the largest organ in human body”: please review the statement and add the bibliographic reference

[Answer]:

We have added reference to the sentence as below:

(line 55) Skeletal muscle is the largest organ in human body which accounts for 30% of body weight in women and greater than 40% of body weight in men [1,2].

References:

1. Janssen I, Heymsfield SB, Wang ZM, Ross R. Skeletal muscle mass and distribution in 468 men and women aged 18-88 yr. J Appl Physiol (1985). 2000;89(1):81-8.

2. Frontera W, Ochala J. Skeletal Muscle: A Brief Review of Structure and Function. Calcified tissue international. 2014;96.

59 “and chronic disease may affect muscle mass and function [2] Muscle mass can be indirectly”: add punctuation mark period

[Answer]: 

Thanks for pointing out the error. We have corrected the mistake as below:

(line 59) and chronic disease may affect muscle mass and function [2].

62 “local muscle endurance [3].” Please verify the reference used

[Answer]: 

Thanks for pointing out the error. We have used Endnote to cite and format the references and make sure their correctness and formatting. The correct references here were as below:

62 “local muscle endurance [5,6].”

5. Jones DA, Rutherford OM, Parker DF. Physiological changes in skeletal muscle as a result of strength training. Q J Exp Physiol. 1989;74(3):233-56.

6. Reid KF, Fielding RA. Skeletal muscle power: a critical determinant of physical functioning in older adults. Exerc Sport Sci Rev. 2012;40(1):4-12.

69 “In general population, a lower HSG is”: Correct HGS

[Answer]: 

We have added “the” prior to general population as suggested by Reviewer 1. We have also done the corrections as below:

(line 69) In the general population, a lower HGS .………..

(line 84) , HGS has also been used to monitor.………..

(line 86) , HGS is a more sensitive indicator to.………..

81, 82 “HGS is one of the recommended tools for assessing nutritional state [13].”: Consider altering to: HGS is one of the four recommended types of measurements to assess nutritional state. [13].

[Answer]: 

We have revised the manuscript as suggested by the reviewer as below:

(line 81-82) HGS is one of the four recommended types of measurements to assess nutritional state.

226 “strsu3ength than R/H in all 3 modes.”: Change to: strength

[Answer]: 

We have corrected the error spelling of strength in line 226. 

258, 259 “Alternatively, segmental BIVA components may be measured with electrodes placed on the ipsilateral arm and foot, and voltage electrodes on both hands.”: Please add reference

[Answer]: 

We have added references to the sentence below.

(line 258-259) Alternatively, segmental BIVA components may be measured with electrodes placed on the ipsilateral arm and foot, and voltage electrodes on both hands [24, 46], as done in our study. 

24. Kushner RF. Bioelectrical impedance analysis: a review of principles and applications. J Am Coll Nutr. 1992;11(2):199-209.

46. Organ LW, Bradham GB, Gore DT, Lozier SL. Segmental bioelectrical impedance analysis: theory and application of a new technique. J Appl Physiol (1985). 1994;77(1):98-112.

Table 1: verify legend

[Answer]: 

Thanks for pointing out the error. We have corrected the error in the footnote as below: 

HGS, hand grip strength; Z, impedance; R, resistance; Xc, reactance; R/H, resistance standardized for height; Xc/H, reactance standardized for height; PhA, phase angle; Lean, lean body mass; BF%, percentage body fat. Subscript DH, RH, LH, WB, RA, and LA denote dominant head, right hand, left hand, whole body, right arm and left arm, respectively. 

Figure 1: add figure title and verify legend

[Answer]: 

1. Asselin MC, Kriemler S, Chettle DR, Webber CE, Bar-Or O, McNeill FE. Hydration status assessed by multi-frequency bioimpedance analysis. Appl Radiat Isot. 1998;49(5-6):495-7.

2. Gudivaka R, Schoeller D, Kushner RF. Effect of skin temperature on multifrequency bioelectrical impedance analysis. J Appl Physiol (1985). 1996;81(2):838-45.

3. Kushner RF. Bioelectrical impedance analysis: a review of principles and applications. J Am Coll Nutr. 1992;11(2):199-209.

4. Gleichauf CN, Roe DA. The menstrual cycle’s effect on the reliability of bioimpedance measurements for assessing body composition. The American Journal of Clinical Nutrition. 1989;50(5):903-7.

For better demostration of Figure 1, we have revised the figure and legend as below:

Figure 1. Illustration of impedance measurement methods for the respective body parts.

---

## [Decision Letter · Decision Letter 1]

28 May 2020

PONE-D-20-07678R1

Prediction and discrimination of skeletal muscle function by bioelectrical impedance vector analysis using a standing impedance analyzer in healthy Taiwanese adults

PLOS ONE

Dear Dr. Hsieh,

Thank you for submitting your manuscript to PLOS ONE. After careful consideration, we feel that it has merit but does not fully meet PLOS ONE’s publication criteria as it currently stands. Therefore, we invite you to submit a revised version of the manuscript that addresses the points raised during the review process.

Prior to the final recommendation for publication of the manuscript, it is necessary for the authors to address the observation made by reviewer #2 in regard to possible incorrectly incorporated bibliographical information.

We look forward to receiving your revised manuscript.

Kind regards,

Jose M. Moran

Academic Editor

PLOS ONE

Reviewers' comments:

Reviewer's Responses to Questions

**Comments to the Author**

1. If the authors have adequately addressed your comments raised in a previous round of review and you feel that this manuscript is now acceptable for publication, you may indicate that here to bypass the “Comments to the Author” section, enter your conflict of interest statement in the “Confidential to Editor” section, and submit your "Accept" recommendation.

Reviewer #1: All comments have been addressed

Reviewer #2: (No Response)

2. Is the manuscript technically sound, and do the data support the conclusions?

Reviewer #1: Yes

Reviewer #2: Yes

3. Has the statistical analysis been performed appropriately and rigorously? 

Reviewer #1: Yes

Reviewer #2: Yes

4. Have the authors made all data underlying the findings in their manuscript fully available?

Reviewer #1: Yes

Reviewer #2: Yes

5. Is the manuscript presented in an intelligible fashion and written in standard English?

Reviewer #1: Yes

Reviewer #2: Yes

6. Review Comments to the Author

Reviewer #1: (No Response)

Reviewer #2: Most of my concerns have been adequately addressed.

I ask you to review Line 55 “Skeletal muscle is the largest organ in human body which accounts for 30% of body weight in women and greater than 40% of body weight in men [1,2].” None of the cited references defines skeletal muscle an organ as it is a tissue so please change it accordingly.

7. PLOS authors have the option to publish the peer review history of their article (what does this mean?). If published, this will include your full peer review and any attached files.

Reviewer #1: No

Reviewer #2: Yes: Maria António Castro

---

## [Author Response · Author response to Decision Letter 1]

31 May 2020

Comments from reviewers:

Reviewer #2

Most of my concerns have been adequately addressed.

I ask you to review Line 55 “Skeletal muscle is the largest organ in human body which accounts for 30% of body weight in women and greater than 40% of body weight in men [1,2].” None of the cited references defines skeletal muscle an organ as it is a tissue so please change it accordingly.

[Answer]: 

We thank the reviewer for her careful and professional review. We agreed with the reviewer that skeletal muscle is regarded as a tissue type instead of an organ in both original information as well as textbooks. Therefore, in our revised manuscript, the word “organ” has been replaced by “tissue”. Please see below for the change in the revised manuscript (lines 56-57): 

Skeletal muscle is the largest tissue in human body which accounts for 30% of body weight in women and greater than 40% of body weight in men [1,2].

---

## [Editor Report · Decision Letter 2]

2 Jun 2020

Prediction and discrimination of skeletal muscle function by bioelectrical impedance vector analysis using a standing impedance analyzer in healthy Taiwanese adults

PONE-D-20-07678R2

Dear Dr. Hsieh,

We are pleased to inform you that your manuscript has been judged scientifically suitable for publication and will be formally accepted for publication once it complies with all outstanding technical requirements.

With kind regards,

Jose M. Moran

Academic Editor

PLOS ONE
---

## [Editor Report · Acceptance letter]

4 Jun 2020

PONE-D-20-07678R2 

Prediction and discrimination of skeletal muscle function by bioelectrical impedance vector analysis using a standing impedance analyzer in healthy Taiwanese adults 

Dear Dr. Hsieh:

I'm pleased to inform you that your manuscript has been deemed suitable for publication in PLOS ONE. Congratulations! Your manuscript is now with our production department. 

Kind regards, 

on behalf of

Dr. Jose M. Moran 

Academic Editor

PLOS ONE